# A novel spectral analysis-based grading system for gastrointestinal activity

Çağlar Cengizler[ID]*

Biomedical Device Technology Program, Vocational School of Health Services, Izmir Democracy University, Izmir, Turkey

* caglar.cengizler@idu.edu.tr

## Abstract

Intestinal sounds, primarily generated by the movement of digested gas and liquids during peristalsis, are acoustic signals that provide valuable insights into intestinal functioning. Traditionally, doctors have relied on stethoscopes to assess the degree of gastrointestinal activity. Recent advancements in computer-aided technologies and electronic stethoscopes have enhanced the understanding and analysis of these sounds. Studies utilizing advanced techniques like deep learning and convolutional neural networks have shown promise in analyzing bowel sounds. Nevertheless, the reliance on personal judgment and the need for large labeled datasets limit the broader applicability of these methods. This study introduces an innovative, unsupervised grading system to objectively evaluate gastrointestinal motility by analyzing bowel sounds through spectral feature analysis. This system offers a practical alternative to traditional listening techniques or complex models. It computes an activity score for digital audio using a cost-effective numerical grading method to assist doctors in quantifying gastrointestinal motility. The method's reliability, validated by Spearman's rank correlation, confirms its accuracy in assessing activity levels and highlights its potential as a reliable and practical tool for supporting objective medical assessments of bowel activity.

## 1 Introduction

Intestinal sounds are acoustic projections from the internals created while the intestine performs its natural functions. Bowel sounds are heard in the abdominal area as rumbling or gurgling [1]. The primary source of these sounds is the movement of digested food, gas, and liquids during intestinal peristalsis [2]. These sounds are directly linked to bowel movements and would reflect valuable information about the intensity of activity and function of the intestine. Therefore, it might be possible to suggest that bowel sounds can be considered a vital sign similar to the sound of a heart [3]. The literature shows that the ability to diagnose changes in the characteristics of gastrointestinal sounds is expanding the perspective on human intestinal physiology [4]. At this point, the stethoscope is the primary tool for listening to and assessing sounds in gastrointestinal activity [5]. With the advancement of technology, electronic stethoscopes and computer-aided systems have also been used. These modern tools make understanding the crucial details of intestinal sounds easier. Many studies show

**Data availability statement:** The first dataset is hosted on Kaggle and can be accessed at

https://www.kaggle.com/datasets/robertnowak/bowel-sounds (accessed on 23/02/2024), under the CC BY-NC 4.0 License. The second dataset is publicly available through the Health Education Media Library at the University of Western Ontario, accessible at https://ir.lib.uwo.ca/clinicalskills_abdominalexam/ (accessed on 16/08/2024), under the Creative Commons Attribution-Noncommercial 4.0 License (CC BY-NC 4.0). The dataset was created by the Arthur Labatt Family School of Nursing and includes audio recordings of bowel sounds. All spectral feature tables used in the analysis have been uploaded as Supporting Information files.

**Funding:** The author(s) received no specific funding for this work.

**Competing interests:** The authors have declared that no competing interests exist.

great potential for computer-aided analysis, which could significantly improve practitioners' ability to distinguish between different bowel conditions [6]. For instance, in one study, Kaneshiro et al. demonstrated the efficiency of bowel sounds in predicting the development of postoperative ileus. They have collected the sounds with a microphone that adheres to the abdominal wall [7]. In another study, gastrointestinal motility was similarly assessed by real-time bowel sound detection. Authors have indicated that their signal power-based approach is promising for detecting the activity [8]. It should be noted that advanced machine learning has been widely used in recent studies to provide robust discrimination ability. In one study, authors have adopted a deep learning approach to propose a smartphone-based instrument that can discriminate bowel sounds [9]. Similarly, another study adapted convolutional neural networks to classify different types of bowel sounds with accurate time stamps. Authors have reported an accuracy of 91.06% [10]. Additionally, Zhang et al. proposed a deep learning mechanism for segmenting bowel sound events, which reportedly outperformed similar approaches. They have implemented a loss function to reduce the sensitivity of their model to thresholds [11]. Long short-term memory is also used to discriminate bowel sounds and noise with MFCC features. Liu et al. reported a classification accuracy of 92.56% with this approach [1]. In addition to deep learning-based methods, some studies have adopted different supervised machine-learning approaches to identify bowel activity frames. For instance, Kölle et al. have implemented support vector machines to discriminate the power distribution over the frequency spectrum [12].

Previous studies have highlighted the necessity of automated bowel sound analysis to support clinical decision-making. While many studies have succeeded using supervised methods, these approaches mainly focus on classification tasks and depend heavily on large labeled datasets. There remains a significant demand for systems capable of providing objective and practical assessments of gastrointestinal activity in clinical settings.

This study addresses this gap by proposing an objective grading system for quantifying the amount of activity based on characteristic changes in the frequency spectrum associated with increased bowel activity. Introduced novel methodology extracts and analyses spectral features to provide a low-cost numerical grading system. It aims to simplify and improve the assessment of bowel activity, particularly in clinical settings where experts may be required to know the gastrointestinal activity levels.

To achieve this, the system calculates the membership of equal-length segments to non-activity regions using several criteria. Multiple features were extracted for evaluation; however, traditional event-based segmentation was not applied. This approach does not require the segmentation of activity regions. The introduced novel methodology aims to simplify and enhance the assessment of bowel activity, allowing for better quantification and insight into bowel activity levels.

## 2 Materials and methods

### 2.1 Data set

In this study, an original data set created by Ficek et al. was utilized (accessed on 23/02/2024). They employed a specialized contact microphone to capture bowel sounds from 19 subjects. A total of 1487 WAV files (44.1kHz, 24bit, two seconds each) from the original data set were analyzed in the study. A 12.5-millisecond Hamming window was chosen to capture rapid changes in bowel sounds. The main goal of this selection was to provide a dynamic assessment of intestinal activity, as the original data authors noted that a window around 10 ms was optimal for their spectral analysis [13]. In addition to the WAV recordings, the authors have provided labels indicating the presence of bowel sounds for each segment, which were used to

calculate the percentage of bowel activity [13]. While these annotations are generally reliable, minor inconsistencies, such as end times exceeding the total signal duration, were observed in some cases. These discrepancies are likely due to manual labeling or slight rounding errors during segmentation. It is important to note that 323 recordings were marked as non-activity, meaning they had no bowel sounds and were used for objective grading. Time domain representations of two different WAV samples belonging to different activity groups are shown in Fig 1.

Additionally, raw bowel sounds from the Health Education Media Library, provided by the Arthur Labatt Family School of Nursing, were utilized for further validation and testing (accessed on 16/08/2024) [14]. The utilized audio was initially sampled at 4kHz and recorded from 4 quadrants of the abdominal region. A total of 8 audio recordings were used, each divided into 83 WAV files after resampling to 44.1kHz. Each WAV file was 2 seconds long. The activity percentage of each WAV file was calculated using empirically placed labels indicating the activity regions, which served as the ground truth. Eight WAV files were detected to contain zero activity and were used as control data. It should be noted that before analysis, all samples from both data sets were initially preprocessed using a bandpass filter with 100 Hz and 1000 Hz cut-off frequencies to suppress irrelevant noise. All data used in this study were anonymized by the original authors before publication and made publicly available under the Creative Commons Attribution-NonCommercial 4.0 International License (CC BY-NC 4.0).

## 2.2 Spectral feature analysis

In this study, all sound segments were characterized using 11 spectral features, and the amount of intestinal activity was graded based on variations in these features. A Hamming window with a 6.25-millisecond overlap was applied along each segment, recalculating the features for each window. The standard deviation of these features across the entire signal was then computed to capture variability and reflect the continuous nature of bowel sounds, providing a holistic assessment of gastrointestinal activity. It should be noted that traditional

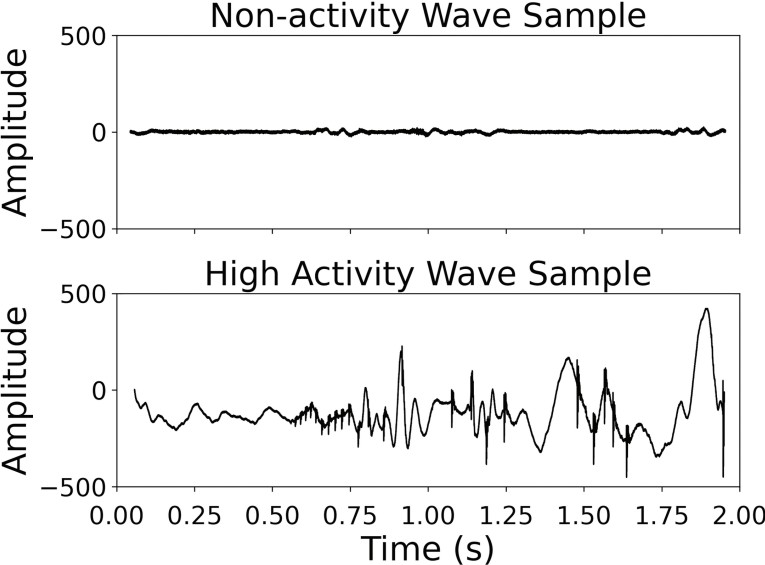

**Fig 1. Representation of a waveform belonging to the non-activity group (top) and high activity group (bottom).**

event-based segmentation was not performed. Instead, each audio recording was divided into equal-length segments of 2 seconds, ensuring consistency in the analysis. The proposed grading system evaluates gastrointestinal motility by analyzing spectral features extracted from these short time windows of bowel sound recordings. Each time window is processed using an unsupervised similarity measure, it assigns a numerical motility score based on its resemblance to a reference non-activity group, consisting of recordings with no bowel activity. The final motility score reflects the similarity between the test window and the control group, providing an objective and quantifiable measure of gastrointestinal function. This system has potential clinical applications in activity monitoring and motility assessment. The experimental setup evaluated six grading metrics to determine the most effective approach: Euclidean Distance, Silhouette Score, Cosine Similarity, KNN Cluster Grade, Cluster Membership Strength, and Gaussian Membership. The workflow of the proposed system is illustrated in Fig 2.

The feature extraction stage was conducted using the MATLAB framework. Eleven spectral features were extracted from each segment utilizing MATLAB's 'audioFeatureExtractor' function. The specific details of the extracted features are provided below.

**Spectral Centroid:** It indicates the centroid of the spectrum, which strongly correlates with the brightness of a sound [15]. The existence of more high-frequency components, likely originating from sharp bowel activities such as rapid muscle contractions, would increase the value of the spectral centroid.

**Spectral Crest:** This feature measures the peakiness of the spectrum. Accordingly, a lower spectral crest may indicate more noise [16]. A higher crest indicates a dominant frequency in bowel sounds originating from specific types of bowel activity, like peristalsis.

**Spectral Decrease:** It would be defined as the rate of spectral amplitude decreases across the spectrum [16]. Softer or rumbling bowel sounds would cause energy to be concentrated in the lower frequencies, resulting in a higher spectral decrease.

**Spectral Entropy:** It measures the complexity of the spectrum. High spectral entropy indicates a more complex or noisy spectrum, while low entropy indicates a purer tone. Moreover, this feature also indicates the amount of information that the signal carries [17]. Irregular contractions in bowel sounds would result in higher entropy, whereas regular, rhythmic sounds would lead to lower entropy.

**Spectral Flatness:** This feature measures how closely a signal resembles either a pure tone or a noise-like sound [18]. A low spectral flatness would indicate increased tonality and a higher presence of tonal components in bowel sounds.

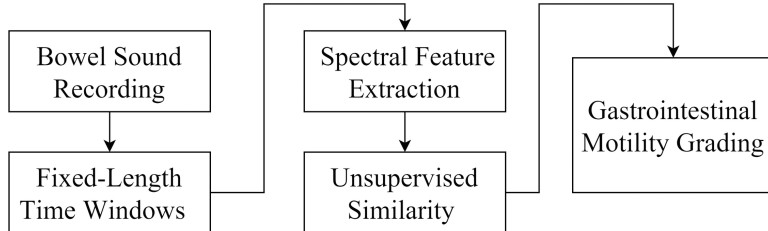

**Fig 2. Flowchart of the proposed system illustrating the grading process based on spectral feature analysis.**

**Spectral Kurtosis:** Spectral kurtosis is a measure that correlates with the shape of the spectral distribution. It evaluates how much a signal's spectrum distribution deviates from a normal distribution, indicating signal impulsiveness [19]. Accordingly, a low kurtosis value would indicate more evenly distributed spectral energy.

**Spectral Roll-off Point:** This is the frequency below which a certain percentage (commonly 85% to 95%) of the total spectral energy is contained. It can be used to measure the spectral shape [15]. A deeper or more resonant bowel may increase the roll-off point value.

**Spectral Skewness:** This feature measures the asymmetry of the spectral shape. Accordingly, it indicates a symmetric structure around the centroid [16]. Deeper bowel sounds with some higher-pitched components would be characterized by positive skewness.

**Spectral Slope:** This feature reflects the amount of decreasing of the spectral amplitude, which would be helpful to measure the tilt of the spectral envelope [20]. Typical low-pitched, rumbling bowel sounds likely to cause a characteristic steep spectral slope, indicating that the sound's energy drops off quickly as frequency increases.

**Spectral Spread:** It is a measure for indicating the spectral variance. It measures the width of spectral distribution, which changes with the amount spread out of the spectrum [16]. A wider spread could indicate the complexity and richness of the sound, reflecting a broad range of frequencies.

**Harmonic Ratio:** This metric measures the energy ratio in the periodic (harmonic) components to the total energy of the sound. It is particularly variable in the unvoiced (noisy) parts of a signal [21]. Accordingly, regular, rhythmic contractions or vibrations would enhance the tonal quality of the sound, thereby increasing the harmonic ratio.

## 2.3 Membership based evaluation

The study aimed to assess the amount of activity in each segment with respect to its resemblance to the non-activity (control) cluster. Accordingly, several membership metrics were calculated and compared to reveal the functionality of the approach. All examined approach's functionality are explained below.

**2.3.1 Euclidean distance.** Euclidean distance is a common metric for measuring the straight-line distance between two points in a multi-dimensional space. Euclidean distance can effectively evaluate an observation's resemblance to a particular cluster in data analysis or machine learning. For each feature, Euclidean distance $d_i$ between the observation and the centroid (mean) of the control features was calculated by:

$$d_i = \sqrt{\sum_{j=1}^{n}(x_{ij} - \bar{x}_j)^2} \tag{1}$$

where $x_{ij}$ represents the observation, $\bar{x}_j$ represents the centroid, and $n$ is the number of features.

**2.3.2 Silhouette score.** The Silhouette score is a distance metric between two observation clusters [22]. Accordingly, it was aimed to calculate as a measure to evaluate how well each sound segment fits into the non-activity group by using mean intra-cluster and mean nearest-cluster distances. Silhouette score $s_i$ is defined by:

$$s = \frac{b - a}{\max(a, b)} \tag{2}$$

Where *a* is the average distance from the test observation to other observations in the same cluster, and *b* is the minimum average distance from the test observation to other observations in a different cluster. This study uses MATLAB's 'evalclusters' function to compute the Silhouette score. It should be noted that a single observation was accepted as a singleton cluster, and its individual Silhouette score was calculated based on its distance to the observations belonging to another cluster. Accordingly, the scores of all other observations, which are part of the larger cluster, are also calculated and contribute to the average. A higher Silhouette score in this context indicates that the observation is more similar to the non-activity group, indicating a lower activity level.

**2.3.3 Cosine similarity.** Cosine similarity grades the similarity of two vectors, based on the cosine of the angle between two vectors [23]. It would be particularly effective in high-dimensional spaces, where Euclidean distances can cause increased computational demand. This study uses cosine similarity to assess how closely an observation aligns with the control cluster regarding directionality and orientation in the vector space. Cosine Similarity $c_i$ was calculated by:

$$c_i = \frac{x_i \cdot \bar{x}}{\|x_i\| \cdot \|\bar{x}\|} \tag{3}$$

where $x_i$ represents an individual observation vector and $\|x_i\|$ is the Euclidean norm of $x_i$, and $\|\bar{x}\|$ is the Euclidean norm of $\bar{x}$. Thus, if an observation is oriented in a similar direction to the control cluster, it receives a higher cosine similarity score.

**2.3.4 K-nearest neighbors (KNN) cluster grade.** This measure was calculated by identification of the *k* closest members of the control cluster to a particular observation [24]. Accordingly, overall grading was based on local rather than global structure. The metric, denoted as *g*, is determined by:

$$g = \frac{1}{k} \sum_{i=1}^{k} d(\text{observation}, \text{neighbor}_i) \tag{4}$$

where $d(\text{observation}, \text{neighbor}_i)$ calculates the Euclidean distance between the observation and its $i^{th}$ nearest neighbour within the cluster. Here, *k* is the number of closest neighbors, which allows us to adjust how we judge whether an observation fits well with the control cluster. Accordingly, a smaller average distance means the observation is closer to the cluster's center, indicating a better fit. It should be noted that the parameter k was accepted as a fine-tuning hyper-parameter and determined through empirical testing to find the optimal number.

**2.3.5 Cluster membership strength.** The Cluster Membership Strength depends on the average distance to all other control observations. Accordingly, the metric computation was based on the mean of the Euclidean distance between the observation in question and each member of the control cluster. Unlike the K-Nearest Neighbors (KNN) Cluster Grade, which considers only the *k* nearest members, the CMS evaluates the observation's relationship with all control observations. This metric was calculated as the inverse of the average Euclidean distance between the observation and each control cluster member. The CMS is defined as follows:

$$CMS = \frac{1}{\frac{1}{N} \sum_{i=1}^{N} d(\text{observation}, \text{member}_i) + eps} \tag{5}$$

Here, $d(\text{observation}, \text{member}_i)$ computes the Euclidean distance between the observation and the $i^{th}$ member of the control cluster, and *N* represents the total number of members in the control cluster. A minimal value (*eps*) is added to the average distance to prevent division

by zero. This approach grades lower average distance with higher membership scores, which indicates a stronger association with the control cluster.

**2.3.6 Gaussian membership.** Gaussian membership is one of the fuzzy membership functions. It grades the degree of resemblance of an observation to a particular cluster based on the assumption that data in each cluster follows a Gaussian distribution. Accordingly, Gaussian membership $m$ was calculated based on a Gaussian distribution model, which represents the probability of the test observation belonging to the control cluster:

$$m = \exp\left(-\frac{(x_i - \bar{x})^2}{2\sigma^2}\right) \tag{6}$$

where $\sigma$ represents the standard deviation of the Gaussian distribution [25]. The output is between 0 and 1, while a higher score indicates a higher degree of membership to the control cluster.

## 2.4 Evaluation of the performance

Performance evaluation was based on an objective comparison between the ground truth and the final scores of each metric. All test recordings were analyzed and ranked according to their scores to determine which metrics most closely align with the actual rankings of bowel sound activity. Performance evaluation was thus completed by comparing the actual sequence of observations (based on expert assessments) against the calculated sequence derived from each grading method. This comparison aimed to quantify the extent to which each grading method accurately reflects the actual sequence of observations in terms of their similarity to the control cluster. Spearman's rank correlation coefficient was calculated for objective comparison. It is a statistical measure of the degree of association between two ranked variables (order of the observations in our case). This non-parametric measure could be used efficiently as a metric even without a linear relationship between the variables [26]. The Spearman's rank correlation coefficient, denoted as $\rho$ is computed as follows:

$$\rho = 1 - \frac{6 \sum d_i^2}{n(n^2 - 1)} \tag{7}$$

Where $d_i$ is the difference between the ranks of each observation in the actual order and the calculated order ($n$ is the number of observations). The ranks for the actual and calculated orders were determined based on the magnitude of the measurements from each grading method, with tied ranks accounted for in cases of identical values. The coefficient values range from -1 to 1, where 1 indicates a perfect positive correlation, -1 indicates a perfect negative correlation, and 0 indicates no correlation. A higher positive value of $\rho$ was accepted as an indicator of the grading method's effectiveness in accurately ranking observations according to their similarity to the control cluster, thus validating the method's performance.

**2.4.1 Feature selection.** A greedy Forward Selection (GFS) approach was implemented for feature selection in the study. The GFS method starts with an empty set of features and iteratively adds features that maximize the model's performance, one at a time, until no further improvements can be achieved [27]. The contribution of remaining features was measured individually with each iteration by temporarily including them in the feature set and assessing the model's performance. The measurement was based on comparing the orders of the observations graded with Spearman's rank correlation, reflecting the predictive power and relevance of the included features.

# 3 Results

The experimental set-up was implemented to evaluate the effectiveness of the membership-based bowel activity scoring approach. Each membership criterion was examined using test observations, with the previously marked activity labels accepted as ground truth. The amount of activity was initially calculated for each WAV file, which was then sorted based on the level of activity they contained. The actual order of observations reflects the ranking of sound segments according to the amount of bowel activity present. Performance grading was completed by comparing the actual and resulting orders of the test observations. Fig 3 illustrates the distribution of bowel activity percentages for both datasets. The horizontal axis represents the percentage of detected bowel activity in each segment, while the vertical axis corresponds to the frequency of occurrences. Dataset 1 consists of 1487 samples, whereas Dataset 2 includes 83 samples. The two distributions highlight potential variations in activity levels across datasets.

Two different data sets were analyzed during the study. Initially, three subsets were formed with the data set 1. Selection and grouping criteria were activity percent. Subset properties are presented in Table 1. It should be noted that control observations are the recordings labeled as zero activity silence segments for both data sets.

A greedy feature selection process is performed with each membership criterion to reveal the best-resulting feature combinations and scores. Each feature combination was initially z-scored before any further analysis. Results are presented in Table 2.

Table 2 indicates that KNN-based scoring is the most effective sorting criterion for all subsets. Accordingly, KNN was additionally examined with all observations between 10% and 50% bowel activity to evaluate its performance at different activity levels. The results are plotted in Fig 4.

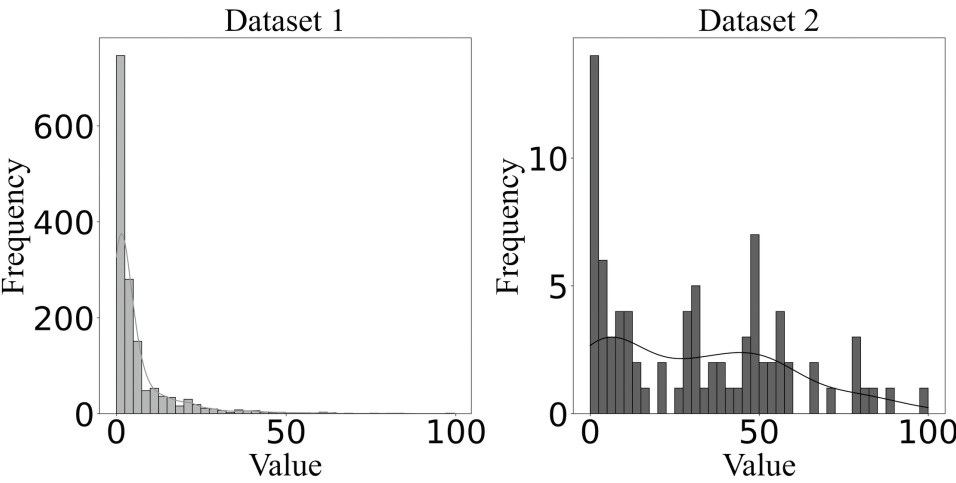

**Fig 3. Histograms showing the distribution of activity percentages for data set 1 (left) and data set 2 (right).**

**Table 1. Properties of subsets used for performance evaluation.**

| Subset | Activity Percent | Number of Test Segments | Number of Control Segments |
|---|---|---|---|
| Subset 1 | >%30 | 53 | 323 |
| Subset 2 | >%40 | 30 | 323 |
| Subset 3 | >%50 | 16 | 323 |

**Table 2. The best resulting feature combinations are presented for all similarity criteria.**

| Subset 1 | | |
| --- | --- | --- |
| **Membership Criteria** | **Best Score** | **Spectral Feature Combination** |
| Euclidean Distance | 0.461 | Skewness, Slope |
| Silhouette Score | 0.435 | Crest, Kurtosis, Skewness, Slope, Harmonic Ratio |
| Cosine Similarity | 0.441 | Crest, Entropy, Flatness, Slope, Harmonic Ratio |
| KNN Score | 0.360 | Harmonic Ratio |
| Cluster Membership | 0.466 | Crest, Skewness |
| Gaussian Membership | 0.461 | Skewness, Slope |
| **Subset 2** | | |
| **Membership Criteria** | **Best Score** | **Spectral Feature Combination** |
| Euclidean Distance | 0.463 | Skewness |
| Silhouette Score | 0.594 | Crest, Kurtosis, Skewness |
| Cosine Similarity | 0.690 | Centroid, Kurtosis, Slope |
| KNN Score | 0.726 | Centroid, Entropy, Kurtosis, Slope |
| Cluster Membership | 0.673 | Crest, Kurtosis, Skewness, Slope |
| Gaussian Membership | 0.463 | Skewness |
| **Subset 3** | | |
| **Membership Criteria** | **Best Score** | **Spectral Feature Combination** |
| Euclidean Distance | 0.544 | Crest, Entropy, Skewness |
| Silhouette Score | 0.688 | Centroid, Crest |
| Cosine Similarity | 0.721 | Centroid, Crest, Slope, Harmonic Ratio |
| KNN Score | 0.874 | Entropy, Slope |
| Cluster Membership | 0.803 | Centroid, Slope |
| Gaussian Membership | 0.544 | Crest, Entropy, Skewness |

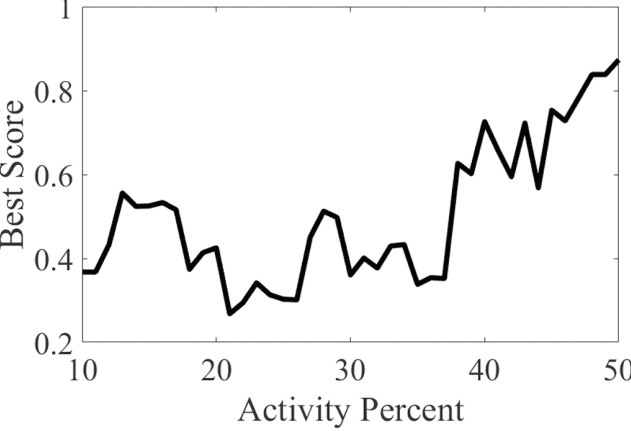

**Fig 4. The effect of increasing bowel activity percent on grading capability of KNN based membership measure is shown.**

To further analyze the effectiveness of membership-based scoring, Fig 5 compares the sorting accuracy of the three best membership criteria. Each segment was color-coded based on its bowel activity level to represent the sorting performance visually. The ground truth represents the actual order of the segments, allowing a direct comparison with the predicted rankings.

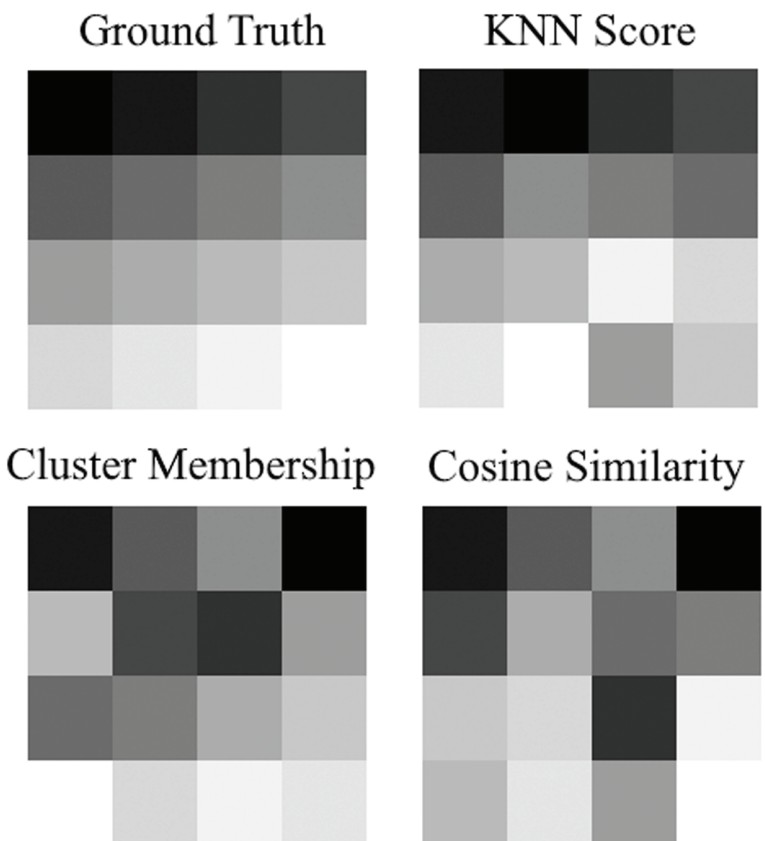

**Fig 5. Visual demonstration of the top three criteria's scoring ability, with segments color-coded by bowel activity level.** The ground truth indicates the actual sequence of segments.

In addition to the analysis with Data set 1, WAV samples from Data set 2 (processed with identical windowing and preprocessing parameters) were also utilized to validate the potential of the presented approach. The scoring ability of the top three criteria was evaluated across different bowel activity levels using the recordings. For each activity level, higher activity percentages were similarly compared with zero-activity samples. Greedy feature selection was conducted at varying activity levels. The findings are illustrated in Fig 6, where the horizontal axis represents the percentage of bowel activity, and the vertical axis shows the Spearman's rank correlation score.

The results indicate that different criteria exhibit varying levels of effectiveness in detecting bowel activity. Cosine similarity demonstrated a moderate relationship with bowel activity, showing improved performance at higher activity levels. In contrast, KNN-based and cluster membership grading provided more stable and reliable correlations across different activity levels. Skewness and slope were consistently selected as the most relevant for activity grading among the extracted features.

It should be noted that each test observation is compared with the control cluster individually in this study. Accordingly, membership criteria based on the comparison of two groups of observations are calculated with a single observation versus the control group or as a singleton cluster over the control cluster. If one data set's control group (non-activity)

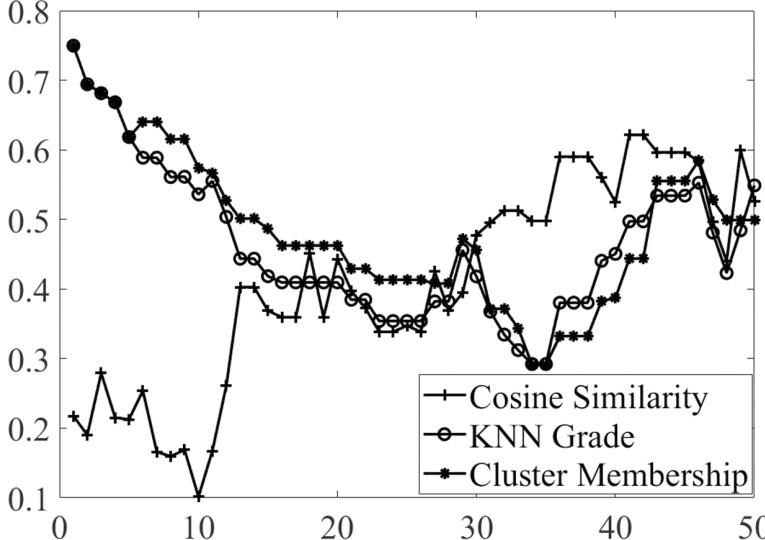

**Fig 6. Validation of the activity grading performance of the top three criteria across different bowel activity levels using data set 2.**

has more variance or internal diversity, it could affect how test observations are scored. During the experiments, it was observed that the grading performance of cosine similarity and silhouette score metrics could be skewed because of the distribution of the control observations. Typically, a higher cosine and silhouette score would indicate more resemblance to the non-activity control group, with lower scores likely pointing to more activity. However, when sorted in descending order, these criteria produced better Spearman rank correlation results (a higher score means more activity) for one data set. In contrast, the opposite sorting order (lower score means more activity) improved results for another data set. Despite the performance differences, a consistent sorting approach was maintained across data sets to ensure uniformity in the analysis.

Furthermore, a comparative evaluation was also performed to demonstrate the proposed method's performance relative to existing approaches. Accordingly, the second data set was first stratified and split into training and test sets to ensure that the distribution of the activity percent was consistent across both sets. Consequently, observations with zero activity were excluded from the stratified sets and reserved as a control group. The unsupervised method was applied directly to the test set, where it calculated the membership of each observation to a non-activity group based on the spectral features.

For comparison, linear regression, Random Forest, Support Vector Regression (SVR), and K-Nearest Neighbors (KNN) learning models are implemented. Prior to model training, the spectral features were standardized using z-scoring. A greedy feature selection process was similarly performed to identify the optimum features for each model. The results of the experiment are presented in Table 3.

## 4 Discussion

This study proposed an alternative unsupervised grading approach for objectively evaluating bowel activity. Accordingly, spectral features characterizing the activity were extracted to

**Table 3. Results of the proposed unsupervised method compared with various supervised approaches using the second data set.**

| Proposed Approach | | |
|---|---|---|
| Criteria | Best Score | Selected Features |
| Cosine Similarity | 0.4182 | Roll-off Point |
| KNN Grade | 0.6429 | Slope |
| Cluster Membership | 0.6453 | Slope |
| Supervised Methods | | |
| Criteria | Best Score | Selected Features |
| Linear Regression | 0.6363 | Slope |
| Random Forest | 0.5680 | Slope, Centroid |
| SVR | 0.6363 | Slope |
| KNN | 0.5753 | Slope |

simplify and effectively assess bowel activity without extensive segmentation. Results show-case significant promise compared to traditional supervised methods. Previous studies under-line that exploiting the advantages of alternative potential spectral features would be neces-sary to increase the current models' performance [2]. One contribution of this study is the identifying the most effective basic spectral features that vary with increased bowel activ-ity. Accordingly, the proposed cluster membership-based scoring system was evaluated with two independent data sets to examine features with related to bowel activity. Extracted fea-tures from both data sets were evaluated via the implemented greedy feature selection scheme. The results indicate that the proposed method is promising; however, it should be noted that applying our system in a clinical setting would require further validation with larger, more diverse data sets to ensure its efficiency.

In this study, instead of discriminating and labeling intestinal activity regions from record-ings, the segments of the recordings were directly processed, and the amount of activity sound they contained was scored. Thus, digital recordings were not subjected to direct classification but were objectively scored by comparison with a group of observations that did not contain activity. This numerical approach could be used as an objective and accurate criterion rather than listening to the intestine with a stethoscope for judgment. It has been reported in the lit-erature that neural network-based classification would require a large amount of data for bet-ter generalization ability. Moreover, such a mechanism's detection sensitivity would be criti-cally affected by noise [1]. Since no discrimination has been made to assign sound segments to a class in this study, it can be considered that the proposed approach could partially over-come some of the challenges (such as over-fitting or sensitivity to noise) that conventional trained machine learning approaches suffer from.

Dividing the audio recordings into equal-length segments creates a consistent basis for comparing different recordings, improving feature extraction reliability and intestinal activity grading. Additionally, using fixed-length segments and predefined spectral features reduces computational load, potentially enabling quicker analysis. These improvements make the method particularly suited for continuous monitoring, where accuracy and speed are critical. In wearable technology scenarios, segmentation might be unavoidable; however, a continuous fixed time interval analysis approach can capture the overall characteristics of bowel sounds while ensuring robust feature representation without biases introduced by incorrect segmen-tation. Therefore, this approach offers a plausible methodology for consistently monitoring and evaluating gastrointestinal motility.

One critical factor for the proposed grading system is the necessity of a standardized recording environment. Even in noisy environments, if the statistical distribution of noise

remains consistent, the cluster-based grading approach can still effectively assess the activity level. However, unexpected noise sources and substandard audio recordings may introduce variability, underscoring the importance of a relatively quiet environment and stable recording conditions. This becomes particularly crucial in wearable and portable device applications, where environmental noise fluctuations could significantly affect grading performance.

It should be noted that, in this study, distinguishing different bowel activities was not the primary aim. While the current method does not specifically distinguish between different gastrointestinal conditions such as irritable bowel syndrome (IBS) or bowel obstruction, future work could extend the approach to identify specific spectral features associated with different disorders using more tailored analysis windows and spectral features.

Additionally, in its current form, the proposed grading system functions as a feature extractor in cases where bowel abnormalities are associated with increased or decreased activity levels. Although the presented methodology is not designed to differentiate between specific gastrointestinal disorders, it still has the potential to support the detection of various pathologies, such as post-operative ileus. Furthermore, the proposed method could be integrated into deep neural networks in future studies, where it may serve as a building block within a novel diagnostic architecture. This integration could enhance the diagnostic capabilities of deep learning models by incorporating the grading system as an intermediate feature extraction layer.

Results with both data sets can be seen as an indication that KNN and cluster membership based scoring might be more effective at distinguishing between varying levels of bowel activity, even in cases where the total amount of activity changes. The diversity in the best-performing spectral feature combinations across different criteria and subsets may have originated from the complex nature of bowel sounds. This underscores the necessity of a nuanced approach to their analysis. Moreover, the imbalance between the number of observations in the test and control groups, particularly when test observations are treated as singleton clusters, can cause metrics like cosine similarity and silhouette score to disproportionately reflect alignment with the larger control group, leading to higher scores that may better correlate with activity levels in one data set but not in another. Additionally, differences in the internal diversity and distribution of observations within the control group across data sets could further explain the variation in results.

Table 2 underscores the effectiveness of various membership criteria across different subsets of bowel activity. The KNN Score was identified as the most effective sorting criterion across all subsets from Data Set 1, particularly in Subset 3, where it achieved the best score of 0.874 using spectral entropy and slope features. This observation is further supported by Fig 5, which provides a visual comparison of the sorting accuracy of the top three membership criteria, highlighting KNN's superior performance.

To further validate the grading performance across different bowel activity levels, Fig 6 presents the results obtained using Data Set 2. The findings indicate that the effectiveness of membership-based scoring varies with activity levels. While Cosine Similarity exhibited a moderate relationship with bowel activity, its performance improved at higher activity levels. In contrast, KNN-based and cluster membership grading showed more stable and consistent correlations across all levels. These results suggest that some criteria are more reliable in distinguishing activity levels and that selecting an appropriate method based on the expected range of activity could enhance grading accuracy.

Additionally, results with data set 2 also highlight the significance of the slope feature, which was consistently selected as the optimal feature by both supervised and unsupervised methods. The discrepancies between these methods suggest that while the slope is dominant, other spectral features, such as the roll-off point and centroid, may also play significant roles

depending on the model used. The proposed unsupervised approach achieved the highest Spearman Rank Correlation of 0.645, slightly outperforming the supervised methods with the limited training data. This suggests that the proposed method is robust across different data sets and has the potential for effectively ranking activity levels. However, the variations in performance across different methods and data sets, as reflected in the differences among the supervised methods, highlight the importance of optimizing the feature space.

It should be noted that the Greedy Feature Selection (GFS) process used in this study is not part of the final application but rather an experimental setup to determine the most relevant features for grading gastrointestinal activity. Once the optimal feature set is identified, there is no need to repeat feature selection in a real-world implementation, minimizing any risk of overfitting in the proposed method. Future studies could explore larger and more diverse datasets with additional spectral features to refine the feature space further. Additionally, alternative feature selection algorithms could be investigated to enhance the robustness of feature selection and improve generalization.

In this study, it was observed that an increased amount of bowel activity causes the segments to differ more from the non-activity group. Accordingly, sound segments with increased bowel activity were scored more accurately (Table 2, Fig 4). Furthermore, the improvement in best scores from Subset 1 to Subset 3 of data set 1, especially noticeable in the Cosine Similarity and Cluster Membership criteria performance, suggests that the grading system's accuracy increases with higher percentages of bowel activity. This trend could have significant implications for diagnosing conditions with increased gastrointestinal motility. A limitation observed is the fluctuation in the effectiveness of different membership criteria across data with varying levels of activity and distribution. The results demonstrate that the best-performing membership criteria can accurately score most segments. However, future studies should focus on improving the proposed approach by integrating additional spectral features and refining the data collection process to enhance diagnostic precision. The timing and methodology of the audio recordings might critically affect performance. In the experiments with the second dataset, non-activity segments were collected from different WAV samples recorded from different quadrants, which may have influenced the grading performance of the proposed scheme. Using the most significant features and a well-matched control group would enhance the accuracy of the grading activity. A standardized approach would also likely improve accuracy. This study highlights the potential of the evaluated spectral features and the variance approach when applied with supervised methods.

However, further investigation is needed to understand how different sorting and feature selection criteria might impact the results, particularly in diverse datasets. Since the proposed approach has not yet been clinically tested, a prospective clinical study on a larger patient population could help validate its effectiveness. Such a study could also explore the interaction between additional spectral features and the influence of model-specific characteristics, refining the results and improving generalizability.

## 5 Conclusions

This study proposes an objective evaluation method for assessing the amount of gastrointestinal activity. The presented method does not need a training stage with limited observations; instead, it utilizes several non-activity recordings for numerical comparison. Instead of detecting activity regions within the sound recordings to assess the amount and existence of gastrointestinal motility, fixed-duration whole sound segments were evaluated without any regional classification. It has been revealed that the standard deviation of basic spectral features may be effective descriptors for increased activity. Thus, using this method with

wearable technology for continuous monitoring could be a practical way to evaluate motility. Therefore, the presented novel approach would contribute to automating diagnostic processes, offering a cost-effective solution with broad implications for gastrointestinal health monitoring.

## Supporting information

**S1 Table.** Spectral features extracted from Dataset 1.
(CSV)

**S2 Table.** Spectral features extracted from Dataset 2.
(CSV)

## Acknowledgments

In this study, ChatGPT-4o was used for language editing and grammatical corrections during the preparation of the manuscript. The authors have carefully reviewed all AI-assisted content and take full responsibility for its accuracy.

## Author contributions

**Conceptualization:** Çağlar Cengizler.

**Data curation:** Çağlar Cengizler.

**Formal analysis:** Çağlar Cengizler.

**Funding acquisition:** Çağlar Cengizler.

**Investigation:** Çağlar Cengizler.

**Methodology:** Çağlar Cengizler.

**Project administration:** Çağlar Cengizler.

**Resources:** Çağlar Cengizler.

**Software:** Çağlar Cengizler.

**Supervision:** Çağlar Cengizler.

**Validation:** Çağlar Cengizler.

**Visualization:** Çağlar Cengizler.

**Writing – original draft:** Çağlar Cengizler.

**Writing – review & editing:** Çağlar Cengizler.

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
