## [Decision Letter · Decision Letter 0]

26 Feb 2025

PONE-D-25-03395A novel spectral analysis-based grading system for gastrointestinal activityPLOS ONE

Dear Dr. cengizler,

Thank you for submitting your manuscript to PLOS ONE. After careful consideration, we feel that it has merit but does not fully meet PLOS ONE’s publication criteria as it currently stands. Therefore, we invite you to submit a revised version of the manuscript that addresses the points raised during the review process.

We look forward to receiving your revised manuscript.

Kind regards,

Mohammad Amin Fraiwan

Academic Editor

PLOS ONE

Reviewers' comments:

Reviewer's Responses to Questions

**Comments to the Author**

1. Is the manuscript technically sound, and do the data support the conclusions?

Reviewer #1: Yes

Reviewer #2: Partly

2. Has the statistical analysis been performed appropriately and rigorously? 

Reviewer #1: N/A

Reviewer #2: N/A

3. Have the authors made all data underlying the findings in their manuscript fully available?

Reviewer #1: Yes

Reviewer #2: Yes

4. Is the manuscript presented in an intelligible fashion and written in standard English?

Reviewer #1: Yes

Reviewer #2: No

5. Review Comments to the Author

Reviewer #1: The paper presents a novel and practical spectral analysis-based grading system for objectively evaluating gastrointestinal activity, addressing a significant gap in traditional assessment methods. By utilizing an unsupervised approach to analyze bowel sounds, the study reduces reliance on subjective interpretation and extensive labeled datasets, making it more accessible for clinical use. The method demonstrates robust performance across multiple datasets, validated through statistical measures such as Spearman's rank correlation, highlighting its potential to enhance diagnostic accuracy and facilitating continuous monitoring of gastrointestinal health.

Limitations and Review Questions

1. The accuracy of the grading system may be influenced by the quality of audio recordings, which could introduce variability. What measures can be taken to standardize audio recording conditions to minimize variability in the results?

2. The method does not differentiate between various gastrointestinal disorders, limiting its diagnostic utility. How could the authors modify their approach to allow for the identification of specific gastrointestinal conditions based on spectral features?

3. The feature selection process may lead to overfitting, particularly in smaller datasets, affecting the robustness of the model. What strategies can be implemented to ensure that the feature selection process remains robust and prevents overfitting in diverse datasets?

4. While the system shows promise, it has not been tested in real clinical settings, which is essential for practical application. What steps are necessary for the authors to undertake clinical validation of their grading system to ensure its effectiveness in real-world scenarios?

Reviewer #2: In this study, the authors tried to present an unsupervised grading system to evaluate gastrointestinal motility by analyzing spectral feature of bowel sounds. The work is interesting. However, a lot of issues need to be addressed before publication:

1. Which output gives the information of gastrointestinal motility evaluation? please make this clear using figures or block diagram.

2.Figure 3 is not clear. How did you get these two figures? What is the difference between two data sets? What is the value of the horizontal axis?

3. The description and discussion for figure 4 and figure 5 is missing.

4. Figure 6 is not clear. The description and discussion is also missing.

5. English writing need to be improved for clarity.

6. PLOS authors have the option to publish the peer review history of their article (what does this mean?). If published, this will include your full peer review and any attached files.

Reviewer #1: **Yes: **Asaad Ahmed

Reviewer #2: No

---

## [Author Response · Author response to Decision Letter 1]

24 Mar 2025

Dear Reviewers,

Thank you for the opportunity to submit a revised draft of our manuscript. We greatly appreciate the time and effort that the reviewers dedicated to providing thorough and constructive feedback and insightful comments. We have carefully considered all the reviewer comments and made the necessary revisions to address each of their concerns.

All revisions are highlighted within the manuscript. Below are our responses to the specific reviewer comments:

Reviewer #1:

1. The accuracy of the grading system may be influenced by the quality of audio recordings, which could introduce variability. What measures can be taken to standardize audio recording conditions to minimize variability in the results?

Thanks to the reviewer for raising this important questions. A new paragraph has been added to the Discussion section to address this issue. This update indicates the importance of a standardized recording environment, explains how the system handles consistent noise, and highlights potential challenges from unexpected noise sources. It also underscores the need for stable recording conditions, especially in wearable and portable device applications.

2. The method does not differentiate between various gastrointestinal disorders, limiting its diagnostic utility. How could the authors modify their approach to allow for the identification of specific gastrointestinal conditions based on spectral features?

We thank the reviewer for this valuable comment. To clarify the distinction between the proposed objective grading approach and a classifier for specific gastrointestinal disorders, a new statement has been added to the Discussion section.

Additionally, as the reviewer has kindly pointed out, the current approach does not differentiate between specific gastrointestinal conditions such as IBS or bowel obstruction. However, we acknowledge its potential to support the detection of activity-related pathologies, such as post-operative ileus.

We have also noted that future research could explore the integration of the proposed system into deep learning models, where it may serve as a feature extraction module within more advanced diagnostic frameworks.

3. The feature selection process may lead to overfitting, particularly in smaller datasets, affecting the robustness of the model. What strategies can be implemented to ensure that the feature selection process remains robust and prevents overfitting in diverse datasets?

We sincerely thank the reviewer for this valuable comment. The Greedy Feature Selection (GFS) process in our study is not intended for real-world applications but rather serves as an experimental setup to identify the most relevant features for grading gastrointestinal activity. Once the optimal feature set is determined, there is no need to perform feature selection repeatedly, which minimizes any risk of overfitting in the proposed method. To clarify this point, we have updated the Discussion section. We have also noted that future studies could explore larger datasets and alternative feature selection methods to further improve robustness.

4. While the system shows promise, it has not been tested in real clinical settings, which is essential for practical application. What steps are necessary for the authors to undertake clinical validation of their grading system to ensure its effectiveness in real-world scenarios?

While our study demonstrates the effectiveness of the proposed grading system, it has not yet been clinically tested. We acknowledge the importance of clinical validation and have added a statement in the Discussion section addressing this as the reviewer kindly asked. Specifically, we suggest that a prospective clinical study on a larger patient population could be conducted to validate the system’s effectiveness. Such a study could also explore the interaction between additional spectral features and model-specific characteristics, further refining the results and improving generalizability.

Reviewer #2:

1. Which output gives the information of gastrointestinal motility evaluation? Please make this clear using figures or block diagram.

Thanks to the reviewer for this important suggestion. To improve clarity, we have revised the manuscript by describing the grading process in the Spectral Feature Analysis subsection. Additionally, we have updated the corresponding figure to illustrate the overall workflow of the grading system. The revised figure is now included as Figure 2 in the manuscript.

2. Figure 3 is not clear. How did you get these two figures? What is the difference between two data sets? What is the value of the horizontal axis?

Thanks to the reviewer for valuable feedback. We have revised the Results section to clarify the differences between the two datasets. We have also noted that the horizontal axis represents bowel activity percentages, while the vertical axis shows their frequency.

3. The description and discussion for figure 4 and figure 5 is missing.

Thanks to the reviewer for this important feedback. The descriptions of Figures 4 and 5 have been revised for better clarity as suggested. Additionally, the incorrect figure reference (Figure 1 instead of Figure 4) has been corrected in the Discussion section. The Discussion now clearly describes Figures 4 and 5, explicitly linking Figure 5 to the evaluation of different membership criteria and highlighting KNN’s performance.

4. Figure 6 is not clear. The description and discussion is also missing.

We sincerely thank the reviewer for this valuable feedback. We have revised the results section to clarify the description of Figure 6. Also, we have explicitly stated that Figure 6 illustrates the grading performance of the top three membership criteria across different bowel activity levels using Data Set 2. The axes have been clearly defined, and the findings have been explained to highlight the trends observed. Additionally, a new paragraph has been added to the Discussion section to further elaborate on the implications of Figure 6.

5. English writing need to be improved for clarity.

Thanks to the reviewer. A general English revision is performed for language refinement.

---

## [Decision Letter · Decision Letter 1]

9 Apr 2025

A novel spectral analysis-based grading system for gastrointestinal activity

PONE-D-25-03395R1

Dear Dr. cengizler,

We’re pleased to inform you that your manuscript has been judged scientifically suitable for publication and will be formally accepted for publication once it meets all outstanding technical requirements.

Kind regards,

Mohammad Amin Fraiwan

Academic Editor

PLOS ONE

Additional Editor Comments (optional):

Reviewers' comments:

Reviewer's Responses to Questions

**Comments to the Author**

1. If the authors have adequately addressed your comments raised in a previous round of review and you feel that this manuscript is now acceptable for publication, you may indicate that here to bypass the “Comments to the Author” section, enter your conflict of interest statement in the “Confidential to Editor” section, and submit your "Accept" recommendation.

Reviewer #1: All comments have been addressed

Reviewer #2: All comments have been addressed

2. Is the manuscript technically sound, and do the data support the conclusions?

Reviewer #1: Yes

Reviewer #2: Yes

3. Has the statistical analysis been performed appropriately and rigorously? 

Reviewer #1: Yes

Reviewer #2: Yes

4. Have the authors made all data underlying the findings in their manuscript fully available?

Reviewer #1: Yes

Reviewer #2: Yes

5. Is the manuscript presented in an intelligible fashion and written in standard English?

Reviewer #1: Yes

Reviewer #2: Yes

6. Review Comments to the Author

Reviewer #1: thanks, all are done

there are not any additional comments.

the paper is reviewed by authors

well, done

Reviewer #2: The authors have addressed all issues that I suggested. It can be accepted for publication in current form.

7. PLOS authors have the option to publish the peer review history of their article (what does this mean?). If published, this will include your full peer review and any attached files.

Reviewer #1: **Yes: **Ahmed A. A. Gad-Elrab

Reviewer #2: **Yes: **Yuee Li

---

## [Editor Report · Acceptance letter]

PONE-D-25-03395R1

PLOS ONE

Dear Dr. Cengizler,

I'm pleased to inform you that your manuscript has been deemed suitable for publication in PLOS ONE. Congratulations! Your manuscript is now being handed over to our production team.

Kind regards,

on behalf of

Dr. Mohammad Amin Fraiwan

Academic Editor

PLOS ONE